# A new integrable structure associated to the Camassa–Holm peakons

J. Avan[(a)], L. Frappat[(b)], E. Ragoucy[(b)]* [1]

**(a)** Laboratoire de Physique Théorique et Modélisation,
CY Cergy Paris Université, CNRS, F-95302 Cergy-Pontoise, France
**(b)** Laboratoire d'Annecy-le-Vieux de Physique Théorique LAPTh,
CNRS, Université Savoie Mont Blanc, F-74940 Annecy

November 17, 2023

## Abstract

We provide a closed Poisson algebra involving the Ragnisco–Bruschi generalization of peakon dynamics in the Camassa–Holm shallow-water equation. This algebra is generated by three independent matrices. From this presentation, we propose a one-parameter integrable extension of their structure. It leads to a new $N$-body peakon solution to the Camassa–Holm shallow-water equation depending on two parameters.

We present two explicit constructions of a (non-dynamical) $r$-matrix formulation for this new Poisson algebra. The first one relies on a tensorization of the $N$-dimensional auxiliary space by a 4-dimensional space. We identify a family of Poisson commuting quantities in this framework, including the original ones. This leads us to constructing a second formulation identified as a spectral parameter representation.

## Contents

[1]avan@cyu.fr, luc.frappat@lapth.cnrs.fr, eric.ragoucy@lapth.cnrs.fr

# 1 Introduction

Many non-linear two-dimensional $(x, t)$ integrable fluid equations exhibit so-called peakon solutions which take the generic form

$$u(x, t) = \sum_{i=1}^{N} p_i(t)\, e^{-|x - q_i(t)|}\,. \tag{1.1}$$

Their dynamics for $(p_i, q_i)$ is deduced from a reduction of the 1+1 fluid equations for $u(x, t)$ and may have integrability properties. The best known example of such integrability feature for peakon equations is given by the Camassa–Holm shallow-water equation [5,6]

$$u_t - u_{xxt} + 3uu_x = 2u_x u_{xx} + uu_{xxx}\,. \tag{1.2}$$

Integrability of the peakons themselves was studied in particular in [7]. A one-parameter extension was proposed by Ragnisco and Bruschi [8], who proved integrability from an implicit construction of a dynamical $r$-matrix. An explicit construction of this dynamical $r$-matrix is still lacking. In this paper, we propose an alternative construction of the Poisson structure relevant for integrability. It relies on a non-dynamical $r$-matrix formulation and uses three dynamical Lax-type matrices, instead of the two matrices introduced by Ragnisco and Bruschi. A direct consequence is that the new integrable peakon model depends on two coupling constants. It provides a more general peakon dynamics which in turns yields a new peakon-type solution of the Camassa–Holm equation (1.2).

We present two representations of the non-dynamical $r$-matrix formulation. The first one requires the use of a larger auxiliary space obtained by tensoring the initial $N$-dimensional space by a 4-dimensional space. One advantage of this representation is that the Yang–Baxter equation for the $r$-matrix structure takes a remarkably simple and compact form. The second representation involves the introduction of a spectral parameter in the $N$-dimensional auxiliary space. Both formulations provide an algebraic framework to construct the same hierarchy of conserved quantities.

The plan of the paper runs as follows.

Section 2 describes the complete Poisson algebra of the three relevant matrix generators involved in our formulation of the integrability properties. It allows the construction of a family of Poisson commuting quantities, including a two-parameter generalization of the peakon Hamiltonian. From the $N$-body dynamics triggered by this new Hamiltonian we deduce a new peakon solution of the Camassa–Holm equation.

Section 3 is devoted to the construction of the $r$-matrix in the extended auxiliary space picture. The explicit computation of the classical Yang–Baxter equation is performed. It is in fact a modified classical Yang–Baxter equation, where the r.h.s. is built from the three-fold tensored Casimir operator of the algebra.

Finally, Section 4 displays the construction of Poisson commuting quantities. It is first done in the framework of the extended auxiliary space. The existence of families of Poisson commuting quantities naturally induces the existence of an alternative, spectral parameter presentation.

# 2  Poisson structure

## 2.1  Description of the original model and its first generalisation

The $n$-peakon solutions (1.1) of the Camassa–Holm equation yield a dynamical system for $p_i, q_i$:

$$\dot{q}_i = \sum_{j=1}^{N} p_j\, e^{-|q_i - q_j|}\,, \qquad \dot{p}_i = \sum_{j=1}^{N} p_i p_j\, \mathfrak{s}_{ij}\, e^{-|q_i - q_j|}\,, \tag{2.1}$$

where $\mathfrak{s}_{ij} = \mathrm{sgn}(q_i - q_j)$. This discrete dynamical system is described by a Hamiltonian

$$H_{CH} = \frac{1}{2} \sum_{i,j=1}^{N} p_i p_j\, e^{-|q_i - q_j|} \tag{2.2}$$

such that

$$\dot{f} = \{f\,, H_{CH}\}\,, \tag{2.3}$$

with the canonical Poisson structure:

$$\{p_i, p_j\} = \{q_i, q_j\} = 0\,, \qquad \{q_i, p_j\} = \delta_{ij}\,. \tag{2.4}$$

The dynamics is encoded in the Lax formulation [4,5]

$$\frac{dL}{dt} = [L, M] \tag{2.5}$$

with

$$L = \sum_{i,j=1}^{N} L_{ij} E_{ij}\,, \qquad L_{ij} = \sqrt{p_i p_j}\, e^{-\frac{1}{2}|q_i - q_j|}\,, \tag{2.6}$$

where $E_{ij}$ is the $N \times N$ elementary matrix with 1 at position $(i, j)$ and 0 elsewhere. The Hamiltonian (2.2) is recast as $H_{CH} = \frac{1}{2}\mathrm{Tr}L^2$.

A generalisation of this original integrable model was proposed by Ragnisco–Bruschi [8], with a Lax matrix[2] $L(\rho) = T + \rho S$, $\rho \in \mathbb{R}$, where

$$T = \sum_{i,j} \sqrt{p_i p_j} \cosh \frac{\nu}{2}(q_i - q_j)\, E_{ij}\,, \tag{2.7}$$

$$S = \sum_{i,j} \sqrt{p_i p_j} \sinh \frac{\nu}{2}|q_i - q_j|\, E_{ij}\,. \tag{2.8}$$

---

[2]One may suppose $\rho \in \mathbb{C}$, but the corresponding Hamiltonians are not real anymore.

The proof of integrability relies on the construction of a dynamical $r$-matrix [8]. The Hamiltonian takes the form

$$
\begin{aligned}
H_{RB}(\rho) &= \frac{1}{2}\mathrm{Tr}L(\rho)^2 \\
&= \frac{1}{2}\sum_{i,j=1}^{N} p_i p_j \Big(\frac{\rho^2+1}{2}\cosh\big(\nu|q_i-q_j|\big) + \rho\,\sinh\big(\nu|q_i-q_j|\big) - \frac{\rho^2-1}{2}\Big).
\end{aligned}
\tag{2.9}
$$

One recovers the original Hamiltonian $H_{CH}$ for the values $\rho = -1$, $\nu = 1$ or $\rho = 1$, $\nu = -1$.

## 2.2 General non dynamical Poisson Structure

The description of the full algebraic structure associated to the Poisson brackets (2.4) and the Lax matrix $L(\rho)$ requires the introduction of a third matrix

$$
A = \sum_{i,j}\sqrt{p_i p_j}\sinh\frac{\nu}{2}(q_i-q_j)\,E_{ij}
\tag{2.10}
$$

which allows to close the Poisson structure of $(T, S)$. It reads

$$
\{T_1, T_2\} = \frac{\nu}{8}\big[\Pi + \Pi^t, A_1 - A_2\big]
\tag{2.11}
$$

$$
\{A_1, A_2\} = \frac{\nu}{8}\big[\Pi - \Pi^t, A_1 - A_2\big]
\tag{2.12}
$$

$$
\{S_1, S_2\} = -\frac{\nu}{4}\big[\Gamma_{12}, S_1\big] + \frac{\nu}{4}\big[\Gamma_{21}, S_2\big] - \frac{\nu}{8}\big[\Pi + \Pi^t, A_1 - A_2\big]
\tag{2.13}
$$

$$
\{T_1, A_2\} = \frac{\nu}{8}\big(\big[\Pi - \Pi^t, T_1\big] - \big[\Pi + \Pi^t, T_2\big]\big)
\tag{2.14}
$$

$$
\{A_1, T_2\} = \frac{\nu}{8}\big(\big[\Pi + \Pi^t, T_1\big] - \big[\Pi - \Pi^t, T_2\big]\big)
\tag{2.15}
$$

$$
\{S_1, T_2\} = \frac{\nu}{4}\big[\Gamma_{21}, T_2\big] \qquad \{T_1, S_2\} = -\frac{\nu}{4}\big[\Gamma_{12}, T_1\big]
\tag{2.16}
$$

$$
\{S_1, A_2\} = \frac{\nu}{4}\big[\Gamma_{21}, A_2\big] \qquad \{A_1, S_2\} = -\frac{\nu}{4}\big[\Gamma_{12}, A_1\big]
\tag{2.17}
$$

with

$$
\begin{aligned}
&\Pi = \sum_{i,j} E_{ij}\otimes E_{ji} \quad;\quad \Pi^t = \sum_{i,j} E_{ij}\otimes E_{ij}\,, \\
&\Gamma_{12} = \sum_{i>j}\Big(E_{ij}\otimes E_{ji} - E_{ji}\otimes E_{ij} + E_{ij}\otimes E_{ij} - E_{ji}\otimes E_{ji}\Big).
\end{aligned}
\tag{2.18}
$$

In (2.11)–(2.17), we have used the auxiliary space description: for any $N \times N$ matrix $M$, we define $M_1 = M \otimes \mathbb{I}_N$ and $M_2 = \mathbb{I}_N \otimes M = \Pi\,M_1\,\Pi$. Similarly, for any matrix $M_{12} \in \mathrm{End}(\mathbb{C}^N) \otimes \mathrm{End}(\mathbb{C}^N)$, we define $M_{21} = \Pi\,M_{12}\,\Pi$.

Remark that, following [2], $\Gamma_{12}$ can be identified with a classical $r$-matrix for the classical open Toda chain.

## 2.3 Generalized peakons

From the above Poisson structure, it is natural to introduce the most general Lax matrix

$$
\bar{L}(\rho, \lambda) = T + \rho S + \lambda A\,,
\tag{2.19}
$$

hereafter denoted $\bar{L}$. The Poisson structure of this Lax matrix reads

$$\{\bar{L}_1, \bar{L}_2\} = -\frac{\nu}{4}\Big(\rho\big[\Gamma_{12}, \bar{L}_1\big] - \rho\big[\Gamma_{21}, \bar{L}_2\big] - \lambda\big[\Pi - \Pi^t, \bar{L}_1 - \bar{L}_2\big] + \lambda\rho\big[\Pi, S_1 - S_2\big]$$
$$- \lambda^2\big[\Pi - \Pi^t, A_2\big] - (1 - \rho^2)\big[\Pi + \Pi^t, A_1\big]\Big). \tag{2.20}$$

We ask the traces $\tau_n = \text{Tr}(\bar{L}^n)$ to be Poisson commuting for all values of $n \in \mathbb{Z}_+$. A direct calculation leads to

$$\{\tau_n, \tau_m\} = \sum_{i=0}^{n-1}\sum_{j=0}^{m-1}\text{Tr}_{12}(\bar{L}_1)^i(\bar{L}_2)^j\{\bar{L}_1, \bar{L}_2\}(\bar{L}_1)^{n-1-i}(\bar{L}_2)^{m-1-j}$$
$$= nm\,\text{Tr}_{12}(\bar{L}_1)^{n-1}(\bar{L}_2)^{m-1}\{\bar{L}_1, \bar{L}_2\}. \tag{2.21}$$

We first notice that for any matrix $M$ we have

$$\text{Tr}_{12}(\bar{L}_1)^{n-1}(\bar{L}_2)^{m-1}\big[\Pi, M_1\big] = \text{Tr}_{12}(\bar{L}_1)^{n-1}(\bar{L}_2)^{m-1}\big(\Pi M_1 - M_1\Pi\big)$$
$$= \text{Tr}_{12}\Big((\bar{L}_1)^{n-1}\Pi(\bar{L}_1)^{m-1}M_1 - (\bar{L}_1)^{n-1}M_1\Pi(\bar{L}_1)^{m-1}\Big)$$
$$= \text{Tr}_1\Big((\bar{L}_1)^{n-1}(\bar{L}_1)^{m-1}M_1 - (\bar{L}_1)^{n-1}M_1(\bar{L}_1)^{m-1}\Big)$$
$$= 0, \tag{2.22}$$

where we have used that for any matrices $U$, $V$, $U_2V_1 = V_1U_2$ and $U_2\Pi = \Pi U_1$ (to get the second line), the property $\text{Tr}_2\Pi = \mathbb{I}_N$ (third line) and the cyclicity of the trace (fourth line). Similarly, we have $\text{Tr}_{12}(\bar{L}_1)^{n-1}(\bar{L}_2)^{m-1}\big[\Pi, M_2\big] = 0$, so that when computing $\{\tau_n, \tau_m\}$, the terms corresponding to $\Pi$ in (2.20) can be dropped. Thanks to this property, we get

$$\{\tau_n, \tau_m\} = -\frac{nm\nu}{4}\text{Tr}_{12}(\bar{L}_1)^{n-1}(\bar{L}_2)^{m-1}\Big(\big[r_{12}, \bar{L}_1\big] - \big[r_{21}, \bar{L}_2\big]\Big)$$
$$+ \frac{nm\nu}{4}\text{Tr}_{12}(\bar{L}_1)^{n-1}(\bar{L}_2)^{m-1}\left((1-\rho^2)\big[\Pi^t, A_1\big] - \lambda^2\big[\Pi^t, A_2\big]\right), \tag{2.23}$$

where $r_{12} = \rho\,\Gamma_{12} + \lambda\,\Pi^t$.

Now, for any matrix $M$ we have $\big[\Pi^t, M_2\big] = \big[\Pi^t, M_1^t\big]$ and since $A$ is an antisymmetric matrix, we get

$$\{\tau_n, \tau_m\} = -\frac{nm\nu}{4}\text{Tr}_{12}(\bar{L}_1)^{n-1}(\bar{L}_2)^{m-1}\Big(\big[r_{12}, \bar{L}_1\big] - \big[r_{21}, \bar{L}_2\big] - (1-\rho^2+\lambda^2)\big[\Pi^t, A_1\big]\Big). \tag{2.24}$$

Furthermore, starting from the relations

$$\text{Tr}_{12}(\bar{L}_1)^{n-1}(\bar{L}_2)^{m-1}\big[R_{12}, \bar{L}_1\big] = \frac{1}{n}\text{Tr}_2\Big((\bar{L}_2)^{m-1}\text{Tr}_1\big[R_{12}, \bar{L}_1^n\big]\Big) = 0,$$
$$\text{Tr}_{12}(\bar{L}_1)^{n-1}(\bar{L}_2)^{m-1}\big[R_{21}, \bar{L}_2\big] = 0, \tag{2.25}$$

valid for any matrix $R_{12}$, we get when $R_{12} = \Pi^t = R_{21}$

$$\text{Tr}_{12}(\bar{L}_1)^{n-1}(\bar{L}_2)^{m-1}\big[\Pi^t, \bar{L}_1 - \bar{L}_2\big] = 0. \tag{2.26}$$

Finally, using that

$$\big[\Pi^t, M_1 - M_2\big] = 0 \qquad \text{for any symmetric matrix } M,$$
$$\big[\Pi^t, U_1 - U_2\big] = 2\big[\Pi^t, U_1\big] \qquad \text{for any anti-symmetric matrix } U, \tag{2.27}$$

we deduce that $\left[\Pi^t,\ \bar{L}_1 - \bar{L}_2\right] = 2\lambda\left[\Pi^t,\ A_1\right]$, so that from (2.26) we get

$$\text{Tr}_{12}(\bar{L}_1)^{n-1}(\bar{L}_2)^{m-1}\left[\Pi^t,\ A_1\right] = 0\,. \tag{2.28}$$

Then, (2.24) rewrites as

$$\left\{\tau_n, \tau_m\right\} = -\frac{nm\nu}{4}\text{Tr}_{12}(\bar{L}_1)^{n-1}(\bar{L}_2)^{m-1}\left(\left[r_{12},\ \bar{L}_1\right] - \left[r_{21},\ \bar{L}_2\right]\right) = 0\,. \tag{2.29}$$

Thus, the model associated to $\bar{L}$ defines an integrable double deformation of the original peakon model.

The Hamiltonian corresponding to these new peakons reads

$$
\begin{aligned}
H_{new}(\rho, \lambda) &= \frac{1}{2}\text{Tr}\bar{L}^2 = H_{RB}(\rho) + \frac{1}{2}\lambda^2\sum_{i,j=1}^{N} p_i p_j \sinh^2\left(\frac{\nu}{2}|q_i - q_j|\right) \\
&= \frac{1}{2}\sum_{i,j=1}^{N} p_i p_j\left(\frac{\rho^2 - \lambda^2 + 1}{2}\cosh\left(\nu|q_i - q_j|\right) + \rho\,\sinh\left(\nu|q_i - q_j|\right) - \frac{\rho^2 - \lambda^2 - 1}{2}\right) \\
&= \sum_{i,j=1}^{N} p_i p_j\left(\frac{(\rho+1)^2 - \lambda^2}{4}e^{\nu\,|q_i - q_j|} + \frac{(\rho-1)^2 - \lambda^2}{4}e^{-\nu\,|q_i - q_j|}\right) - \frac{\rho^2 - \lambda^2 - 1}{4}\mathfrak{p}^2\,,
\end{aligned}
\tag{2.30}
$$

where we introduced

$$\mathfrak{p} = \sum_{i=1}^{N} p_i\,. \tag{2.31}$$

We obtain $H_{CH}$ for $\lambda = 0$, $\rho = -1$ and $\nu = 1$, while $H_{RB}(\rho)$ is recovered when $\lambda = 0$.

Note also that for $(\rho + 1)^2 = \lambda^2$ and $\nu = 1$, we get a shifted version of the original peakon model:

$$H_{new}\bigl(\rho, \pm(\rho+1)\bigr)\Big|_{\nu=1} = -\rho\,H_{CH} - \frac{\rho+1}{2}\mathfrak{p}^2\,. \tag{2.32}$$

A similar result holds for $(\rho - 1)^2 = \lambda^2$ and $\nu = -1$.

Remark that a generic value for $\nu$ can be obtained from the cases $\nu = \pm 1$, see section 2.4 below. Hence the above conditions correspond to a condition on $\rho$ and $\lambda$, rather than two conditions on $\rho$, $\lambda$ and $\nu$.

## 2.4   $N$-body solutions of the fluid equation

We establish here a general result on the consistency conditions for peakon-type $N$-body solutions to (a deformation of) the Camassa–Holm equation, including the Hamiltonian evolution of the $N$-body variables. We first consider the case $\nu^2 = 1$, and then show how a generic value for $\nu$ can be obtained from the cases $\nu = \pm 1$.

### A deformed version of the Camassa–Holm equation

We first restrict ourself to the case

$$\nu^2 = 1\,. \tag{2.33}$$

The Hamiltonian $H_{new}(\rho, \lambda)$ given in (2.30), can be rewritten as

$$H = \frac{1}{2} \sum_{i,j=1}^{N} p_i p_j F(|q_i - q_j|),$$ (2.34)

with

$$F(q) = \frac{\rho^2 - \lambda^2 + 1}{2} \cosh(\nu q) + \rho \sinh(\nu q) - \frac{\rho^2 - \lambda^2 - 1}{2}.$$ (2.35)

Note that the function $F$ in (2.35) obey the following differential equation with the initial value conditions

$$F''(x) - F(x) = \frac{\rho^2 - \lambda^2 - 1}{2}, \qquad F(0) = 1, \qquad F'(0) = \nu \rho,$$ (2.36)

where we have used the property $\nu^2 = 1$.

The Hamiltonian $H_{new}(\rho, \lambda)$ describes a time evolution for $p_i, q_i$, given by:

$$\dot{q}_i = \{q_i, H_{new}(\rho, \lambda)\} = \sum_{j=1}^{N} p_j \, F(|q_i - q_j|),$$

$$\dot{p}_i = \{p_i, H_{new}(\rho, \lambda)\} = \sum_{j=1}^{N} p_i p_j \, \mathfrak{s}_{ij} \, F'(|q_i - q_j|),$$ (2.37)

where $\mathfrak{s}_{ij} = \text{sgn}(q_i - q_j)$ as above, and

$$F'(q) = \frac{d}{dq} F(q) = \nu \left( \frac{\rho^2 - \lambda^2 + 1}{2} \sinh(\nu q) + \rho \cosh(\nu q) \right).$$ (2.38)

To recover the time evolutions (2.37) from a fluid equation, we define $u(x, t)$ as

$$u(x, t) = \sum_{i=1}^{N} p_i(t) \, F(-|x - q_i(t)|)$$ (2.39)

which is a direct generalisation of (1.1). Plugging the form (2.39) into the l.h.s. of the differential relation (2.36) and using (2.31), one finds

$$u_t - u_{xxt} + 3u u_x - 2u_x u_{xx} - u u_{xxx} =$$

$$= \sum_{i=1}^{N} \left\{ \mu \, \dot{p}_i + \text{sgn}(x - q_i) p_i \left( \dot{q}_i \left( F'_i - F'''_i \right) + 2\mu \, \mathfrak{p} \, F'_i \right) \right.$$

$$\left. + 2 \, F'(0) \, \delta(x - q_i) \left( \dot{p}_i - \sum_{j=1}^{N} \mathfrak{s}_{ij} p_i p_j F'_{ij} \right) + 2 \, F'(0) \, \delta'(x - q_i) \left( -\dot{q}_i + \sum_{j=1}^{N} p_j F_{ij} \right) p_i \right\},$$ (2.40)

with

$$\mu = \frac{\rho^2 - \lambda^2 - 1}{2}.$$ (2.41)

We have introduced the notation

$$F_i = F(-|x - q_i|) \quad \text{and} \quad F_{ij} = F(-|q_i - q_j|).$$ (2.42)

From the differential equations (2.36) and the equations of motion (2.37), we get

$$u_t - u_{xxt} + 3uu_x - 2u_xu_{xx} - uu_{xxx} = 2\mu\,\mathfrak{p}\sum_{i=1}^{N}\mathrm{sgn}(x-q_i)p_i\,F_i'\,. \qquad (2.43)$$

To obtain this relation, we have used the property (deduced from the equations of motion (2.37)) that $\mathfrak{p}$ defined in (2.31) is a free constant parameter of the model, i.e. $\dot{\mathfrak{p}} = 0$.

Now remarking that $\sum_{i=1}^{N}\mathrm{sgn}(x-q_i)p_iF_i' = u_x$, we find a modification of the Camassa–Holm shallow-water equation

$$u_t - u_{xxt} + 3\,uu_x - 2\,u_xu_{xx} - uu_{xxx} = 2\mu\,\mathfrak{p}\,u_x\,. \qquad (2.44)$$

For $\mu = 0$ we recover the undeformed Camassa–Holm shallow-water equation. For $\mu \neq 0$, we find the deformed fluid equation (2.44). It exhibits peakon-type solutions (2.39) with the integrable dynamics (2.30).

**Deformed versus undeformed Camassa–Holm equation**

We now show that this deformed equation is in fact equivalent to the original one. We perform the following change of variables and function:

$$y = x - \mu\,\mathfrak{p}\,t\,, \quad t' = t\,, \quad v = u - \mu\,\mathfrak{p}\,. \qquad (2.45)$$

Starting from the equation (2.44), we get back to the undeformed Camassa–Holm equation:

$$v_t - v_{yyt} + 3\,vv_y - 2\,v_yv_{yy} - vv_{yyy} = 0\,. \qquad (2.46)$$

As a consequence, the expression (2.35) for $F$ provides a new $N$-body solution of the Camassa–Holm equation:

$$u(x,t) = -\mu\,\mathfrak{p} + \sum_{i=1}^{N}p_i(t)\,F\Big(-|x + \mu\,\mathfrak{p}\,t - q_i(t)|\Big)\,. \qquad (2.47)$$

In fact, since $\mathfrak{p}$ is a free constant of motion, we get a one-parameter family of solutions.

**Rescaling by $\nu$**

We wish to show that the case $\nu > 0$ can be obtained from the case $\nu = 1$. We start with the model defined with the value $\nu = 1$, with

$$u_{\nu=1}(x,t) = -\mu\,\mathfrak{p} + \sum_{i=1}^{N}p_i(t)\,F_{\nu=1}(-\,|x + \mu\,\mathfrak{p}\,t - q_i(t)|)\,,$$

$$F_{\nu=1}(q) = \frac{\rho^2 - \lambda^2 + 1}{2}\cosh\big(q\big) + \rho\,\sinh\big(q\big) - \frac{\rho^2 - \lambda^2 - 1}{2}\,, \qquad (2.48)$$

$$H_{\nu=1} = \frac{1}{2}\sum_{i,j=1}^{N}p_ip_jF_{\nu=1}(|q_i - q_j|)\,.$$

We perform the symplectic transformation $\bar{q}_i = \frac{1}{\nu}\,q_i$, $\bar{p}_i = \nu\,p_i$, and at the same time a dilation $\bar{x} = \frac{1}{\nu}\,x$. It is easy to see that the Hamiltonian $H_{\nu=1}$ get a $\frac{1}{\nu^2}$ factor, indicating that we need to define $\bar{t} = \frac{1}{\nu^2}\,t$. Then

$$\bar{u}(\bar{x},\bar{t}) = -\mu\,\bar{\mathfrak{p}} + \sum_{i=1}^{N}\bar{p}_i(t)\,F_{\nu=1}(-\nu\,|\bar{x} + \mu\,\bar{\mathfrak{p}}\,\bar{t} - \bar{q}_i(\bar{t})|) \qquad (2.49)$$

obeys

$$\nu^2 \left( \bar{u}_{\bar{t}} + 3\,\bar{u}\bar{u}_{\bar{x}} \right) = \bar{u}_{\bar{x}\bar{x}\bar{t}} + 2\bar{u}_{\bar{x}}\bar{u}_{\bar{x}\bar{x}} + \bar{u}\bar{u}_{\bar{x}\bar{x}\bar{x}} \,. \tag{2.50}$$

In (2.49), $\bar{q}_i$ and $\bar{p}_i$ are canonical variables whose time evolution is triggered by the Hamiltonian $\bar{H} = \frac{1}{\nu^2}\,H_{\nu=1}$.

Note that to obtain (2.49), we have used the property $|\nu| = \nu$, hence the condition $\nu > 0$. To get the other values of $\nu$, one needs to start from the model with $\nu = -1$ and perform the same transformations.

# 3 $\mathscr{R}$-matrix representation

## 3.1 Extended auxiliary space

Up to now, we have used $N$-dimensional auxiliary spaces denoted by 1 and 2. These spaces will be now labelled $0$ and $0'$ respectively. We introduce two additional 4-dimensional auxiliary spaces labelled 1 and 2, and define the resulting tensored auxiliary spaces $\textsc{i}=(0,1)$ and $\textsc{ii}=(0',2)$. We introduce the following $16N^2 \times 16N^2$ diagonal $r$-matrix and $4N \times 4N$ diagonal Lax matrix

$$\mathscr{R}_{\textsc{i},\textsc{ii}} = \frac{1}{2}(\Pi_{00'} - \Pi_{00'}^{t_0})\,U_{12} + \frac{1}{2}(\Pi_{00'} + \Pi_{00'}^{t_0})\,V_{12} - \Gamma_{00'}\,W_{12}\,, \tag{3.1}$$

$$\mathscr{L}_{\textsc{i}} = A_0\,\mathbb{I}_1 + T_0\,X_1 + S_0\,Y_1, \tag{3.2}$$

where

$$X = \mathrm{diag}(1,-1,0,0)\,, \qquad Y = \mathrm{diag}(0,0,1,-1)\,,$$
$$U_{12} = \frac{\nu}{4}\left( \mathbb{I} \otimes \mathbb{I} + \frac{1}{2}\big(X_1^2 - Y_1^2 - X_2^2 + Y_2^2\big) \right)\,, \tag{3.3}$$
$$V_{12} = \frac{\nu}{4}\big(X_1 X_2 - Y_1 Y_2\big)\,, \quad W_{12} = \frac{\nu}{4}Y_2.$$

In the following we will also write

$$\mathscr{R} = \sum_{i,j=1}^{4} \mathfrak{r}_{00'}^{ij}\,e_{ii} \otimes e_{jj} \quad \text{and} \quad \mathscr{L} = \sum_{i=1}^{4} \ell_0^i\,e_{ii}\,, \tag{3.4}$$

where $e_{ij}$ are the $4 \times 4$ elementary matrices. In (3.4), the superscripts indicate the matrix entries in the $4 \times 4$ auxiliary spaces, while the subscripts show in which $N \times N$ auxiliary space they act on. From the above expressions, we get

$$
\begin{aligned}
&\ell_0^1 = A_0 + T_0\,, \qquad \ell_0^2 = A_0 - T_0\,, \qquad \ell_0^3 = A_0 + S_0\,, \qquad \ell_0^4 = A_0 - S_0\,, \\
&\mathfrak{r}_{00'}^{11} = \mathfrak{r}_{00'}^{22} = \frac{\nu}{4}\Pi_{00'}\,, \qquad \mathfrak{r}_{00'}^{12} = \mathfrak{r}_{00'}^{21} = -\frac{\nu}{4}\Pi_{00'}^{t_0}\,, \\
&\mathfrak{r}_{00'}^{13} = \mathfrak{r}_{00'}^{23} = \frac{\nu}{4}(\Pi_{00'} - \Pi_{00'}^{t_0} - \Gamma_{00'})\,, \qquad \mathfrak{r}_{00'}^{14} = \mathfrak{r}_{00'}^{24} = \frac{\nu}{4}(\Pi_{00'} - \Pi_{00'}^{t_0} + \Gamma_{00'})\,, \\
&\mathfrak{r}_{00'}^{33} = -\frac{\nu}{4}(\Pi_{00'}^{t_0} + \Gamma_{00'})\,, \qquad \mathfrak{r}_{00'}^{34} = \frac{\nu}{4}(\Pi_{00'} + \Gamma_{00'})\,, \\
&\mathfrak{r}_{00'}^{43} = \frac{\nu}{4}(\Pi_{00'} - \Gamma_{00'})\,, \qquad \mathfrak{r}_{00'}^{44} = \frac{\nu}{4}(-\Pi_{00'}^{t_0} + \Gamma_{00'})\,, \\
&\mathfrak{r}_{00'}^{31} = \mathfrak{r}_{00'}^{32} = \mathfrak{r}_{00'}^{41} = \mathfrak{r}_{00'}^{42} = 0\,.
\end{aligned}
\tag{3.5}
$$

## 3.2 Poisson brackets and classical Yang–Baxter relation

The relation

$$\big\{ \mathscr{L}_{\mathrm{I}} \, , \, \mathscr{L}_{\mathrm{II}} \big\} = \big[ \mathscr{R}_{\mathrm{I},\mathrm{II}} \, , \, \mathscr{L}_{\mathrm{I}} \big] - \big[ \mathscr{R}_{\mathrm{II},\mathrm{I}} \, , \, \mathscr{L}_{\mathrm{II}} \big] \tag{3.6}$$

is equivalent to

$$\big\{ \ell_0^i \, , \, \ell_{0'}^j \big\} = \big[ \mathfrak{r}_{00'}^{ij} \, , \, \ell_0^i \big] - \big[ \mathfrak{r}_{0'0}^{ji} \, , \, \ell_{0'}^j \big] \, , \qquad 1 \le i \le j \le 4 \, . \tag{3.7}$$

Using the expressions (3.5), one can check by a direct calculation that these relations are equivalent to the Poisson brackets (2.11)–(2.17).

We have also established that $\mathscr{R}$ obeys a modified Yang–Baxter relation

$$\big[ \mathscr{R}_{\mathrm{I},\mathrm{II}}, \mathscr{R}_{\mathrm{I},\mathrm{III}} + \mathscr{R}_{\mathrm{II},\mathrm{III}} \big] + \big[ \mathscr{R}_{\mathrm{III},\mathrm{II}}, \mathscr{R}_{\mathrm{I},\mathrm{III}} \big] = \mathcal{O}_{\mathrm{I},\mathrm{II},\mathrm{III}} \tag{3.8}$$

which reads in component

$$\big[ \mathfrak{r}_{00'}^{ij}, \mathfrak{r}_{00''}^{ik} + \mathfrak{r}_{0'0''}^{jk} \big] + \big[ \mathfrak{r}_{0''0'}^{kj}, \mathfrak{r}_{00''}^{ik} \big] = \mathfrak{o}_{00'0''}^{ijk} \, . \tag{3.9}$$

Using again the expressions (3.5), we compute the r.h.s. of (3.9). To simplify its presentation, we introduce the operators

$$\Omega_{00'0''} = \omega_{00'0''} - (\omega_{00'0''})^{t_0 t_{0'} t_{0''}} \quad \text{with} \quad \omega_{00'0''} = \sum_{a,b,c} E_{ab} \otimes E_{bc} \otimes E_{ca} \, . \tag{3.10}$$

Then, the only non-vanishing $\mathfrak{o}_{00'0''}^{ijk}$ are given by

$$\mathfrak{o}^{111} = \mathfrak{o}^{222} = \mathfrak{o}^{333} = \mathfrak{o}^{444} = \Omega \, , \qquad \mathfrak{o}^{122} = \mathfrak{o}^{211} = \mathfrak{o}^{344} = \mathfrak{o}^{433} = -\Omega^{t_0} \, ,$$
$$\mathfrak{o}^{112} = \mathfrak{o}^{221} = \mathfrak{o}^{334} = \mathfrak{o}^{433} = -\Omega^{t_{0''}} \, , \qquad \mathfrak{o}^{121} = \mathfrak{o}^{212} = \mathfrak{o}^{343} = \mathfrak{o}^{434} = -\Omega^{t_{0'}} \, , \tag{3.11}$$

where we omitted the subscript $_{00'0''}$ to lighten the writing.

To obtain (3.11), we have used the classical Yang–Baxter equation for $\Gamma$ [1]:

$$\big[ \Gamma_{00'}, \Gamma_{00''} + \Gamma_{0'0''} \big] + \big[ \Gamma_{0''0'} \, , \, \Gamma_{00''} \big] = \Omega - \Omega^{t_0} + \Omega^{t_{0'}} + \Omega^{t_{0''}} \, . \tag{3.12}$$

Remark that the formulas (3.11) imply the following formula for $\mathcal{O}_{\mathrm{I},\mathrm{II},\mathrm{III}}$:

$$\mathcal{O}_{\mathrm{I},\mathrm{II},\mathrm{III}} = \Omega_{0,0',0''} \, \mathcal{H}_{123} - \sum_{s=1}^{3} \Omega_{0,0',0''}^{\widehat{t_s}} \, Ad(Z_s)(\mathcal{H}_{123}) \, , \quad \text{with} \quad \begin{cases} \mathcal{H}_{123} = \displaystyle\sum_{j=1}^{4} e_{jj} \otimes e_{jj} \otimes e_{jj}, \\ Z = e_{12} + e_{21} + e_{34} + e_{43}. \end{cases} \tag{3.13}$$

We have defined $\widehat{t_1} = t_0$, $\widehat{t_2} = t_{0'}$, $\widehat{t_3} = t_{0''}$. The notation $Z_s$ stands for the matrix $Z$ acting in the auxiliary space $s$.

# 4 Conserved quantities

## 4.1 Conserved quantities from the $\mathscr{L}$ operator

Now that we have established a Lax presentation of the Poisson brackets, we can consider the traces

$$\begin{aligned} \mathfrak{t}_n^K &= \operatorname{Tr} \operatorname{tr} \big( K \, \mathscr{L}^n \big) \\ &= k^1 \operatorname{Tr}[(A+T)^n] + k^2 \operatorname{Tr}[(A-T)^n] + k^3 \operatorname{Tr}[(A+S)^n] + k^4 \operatorname{Tr}[(A-S)^n] \end{aligned} \tag{4.1}$$

where $K = \sum_i k^i e_{ii} \otimes \mathbb{I}_N$ is a diagonal $4N \times 4N$ matrix, acting as identity in the spaces $0$ and $0'$. We have denoted "Tr" the trace in the $N$-dimensional space and "tr" the trace in the 4-dimensional space. Since all matrices commute in the 4-dimensional space, it is easy to show that

$$\{\mathfrak{t}_n^K, \mathfrak{t}_m^{K'}\} = 0, \quad \forall n, m. \tag{4.2}$$

As a consequence we have a commuting family of operators generated by

$$\mathrm{Tr}[(A+T)^n], \quad \mathrm{Tr}[(A-T)^n], \quad \mathrm{Tr}[(A+S)^n], \quad \mathrm{Tr}[(A-S)^n]. \tag{4.3}$$

Since $A$ is antisymmetric while $T$ and $S$ are symmetric, we get using $\mathrm{Tr}M = \mathrm{Tr}(M^T)$:

$$\mathrm{Tr}[(A+T)^n] = \mathrm{Tr}[(T-A)^n] \quad \text{and} \quad \mathrm{Tr}[(A+S)^n] = \mathrm{Tr}[(S-A)^n]. \tag{4.4}$$

To get more insight on the two remaining generators, we consider the first chamber

$$i < j \iff q_i < q_j. \tag{4.5}$$

Then, looking at the form of $A$ and $S$, one sees that $A+S$ is a triangular matrix with zeros on the diagonal, so that

$$\mathrm{Tr}[(A+S)^n] = 0. \tag{4.6}$$

Similarly, a simple recursion shows that

$$(A+T)^n = \sum_{i_1,\ldots,i_{n+1}} p_{i_2} \cdots p_{i_n} \sqrt{p_{i_1} p_{i_{n+1}}} \, e^{\frac{\nu}{2}(q_{i_1}-q_{i_{n+1}})} e_{i_1,i_{n+1}}. \tag{4.7}$$

We then get

$$\mathrm{Tr}[(A+T)^n] = \mathfrak{p}^n, \tag{4.8}$$

where $\mathfrak{p}$ is given in (2.31).

Since we obtain the other chambers through permutation of rows and columns, the properties (4.6) and (4.8) are valid everywhere. In conclusion, the quantities $\mathfrak{t}_n^K$ provide only one conserved quantity $\mathfrak{p}$ and its corresponding polynomial algebra.

## 4.2 Adding more conserved quantities

It is easy to show that since $2T = \ell^1 - \ell^2$, $2S = \ell^3 - \ell^4$, and $2A = \ell^1 + \ell^2 = \ell^3 + \ell^4$, we have

$$\bar{L} = \frac{1}{2}\Big((1+\delta)\ell^1 + (-1+\delta)\ell^2 + (\rho+\lambda-\delta)\ell^3 + (-\rho+\lambda-\delta)\ell^4\Big)$$
$$= \mathrm{tr}(D\mathscr{L}) \quad \text{with} \quad D = \frac{1}{2}\mathrm{diag}\big(1+\delta, \delta-1, \lambda+\rho-\delta, \lambda-\rho-\delta\big), \tag{4.9}$$

which is valid for any $\delta \in \mathbb{C}$. Then, one recovers the conserved quantities $\tau_n$ as

$$\tau_n = \mathrm{Tr}\Big(\big(\mathrm{tr}\, D\,\mathscr{L}\big)^n\Big). \tag{4.10}$$

Note that to prove this property, we need to consider $\big(\mathrm{tr}D\mathscr{L}\big)^n$ instead of $\mathscr{L}^n$, which explains why the "natural" approach using the elements $\mathfrak{t}_n^K$ does not lead to the full set of conserved quantities.

Remark that when $D$ is replaced by a general diagonal matrix $K$, the set

$$\mathcal{S}_K = \{\tau_n^K, \quad n = 0, 1, 2...\} \quad \text{with} \quad \tau_n^K = \text{Tr}\Big(\big(\text{tr}\, K\, \mathcal{L}\big)^n\Big) \tag{4.11}$$

is still commutative: $\{\tau_n^K, \tau_m^K\} = 0$.

We now consider two diagonal matrices $K$ and $K'$ and look for the vanishing of the Poisson brackets $\{\tau_n^{K'}, \tau_m^K\}$. Using a formal computation software, we are led to propose a family of matrices

$$K(x) = diag\Big( -\frac{\rho - \lambda - 1}{4\rho}\frac{\beta_1(x)}{x+1}, \ -\frac{\rho - \lambda + 1}{4\rho}\frac{\beta_2(x)}{x+1}, \ \frac{\beta_1(x)\beta_2(x)}{8\rho(x+1)}, \ 0\Big) \tag{4.12}$$
$$\beta_1(x) = (\rho + \lambda - 1)x + 3\rho - \lambda + 1 \quad \text{and} \quad \beta_2(x) = (\rho + \lambda + 1)x + 3\rho - \lambda - 1$$

depending on a parameter $x$, such that :

$$\{\tau_n(x), \tau_m(y)\} = 0, \quad \forall x, y \quad \text{with} \quad \tau_n(x) = \text{Tr}\Big(\big(\text{tr}K(x)\,\mathcal{L}\big)^n\Big). \tag{4.13}$$

Note that there are other choices of $K$ matrices leading to the same hierarchy of conserved quantities $\tau_n(x)$. The analytical proof of this property is given at the end of section 4.3. We first provide a spectral parameter formulation of the Poisson brackets.

## 4.3 A spectral parameter representation of the Poisson structure

For fixed $\lambda$ and $\rho$, we introduce a family extending the $\bar{L}$ operator (2.19)

$$\bar{L}(a, \bar{a}) = T + \bar{a}\, S + a\, A \quad \text{with} \quad \frac{a^2 - \bar{a}^2 - 1}{\bar{a}} = \frac{\lambda^2 - \rho^2 - 1}{\rho}. \tag{4.14}$$

Indeed one can check from (4.12) that we have

$$\text{tr}K(x)\,\mathcal{L} = T + \bar{a}(x)\, S + a(x)A \equiv \bar{L}(x) \quad \text{with} \quad \begin{cases} \bar{a}(x) = \dfrac{\beta_1(x)\beta_2(x)}{8\rho(x+1)}, \\[2mm] a(x) = (x+1)\dfrac{(\lambda + \rho)^2 - 1}{8\rho} - \dfrac{(\lambda - \rho)^2 - 1}{2\rho(x+1)}. \end{cases} \tag{4.15}$$

The relation in (4.14) follows from the explicit form of $\beta_1(x)$ and $\beta_2(x)$. Some specific values for $x$ provide interesting sub-cases:

$$x_0 = 1 \qquad\qquad\qquad\qquad \Rightarrow \quad \bar{L}(x_0) = T + \lambda A + \rho S,$$
$$x_\pm = -1 \pm 2\sqrt{\frac{(\lambda - \rho)^2 - 1}{(\lambda + \rho)^2 - 1}} \qquad \Rightarrow \quad \bar{L}(x_\pm) = T + \rho_\pm S, \quad \rho_\pm = \bar{a}(x_\pm), \tag{4.16}$$
$$x_j = -\frac{3\rho - \lambda - (-1)^j}{\rho + \lambda + (-1)^j} \qquad \Rightarrow \quad \bar{L}(x_j) = T + \lambda_j A, \quad \lambda_j = a(x_j), \ j = 1, 2.$$

$\bar{L}(x_0)$ corresponds to the original $\bar{L}$ matrix, while $\bar{L}(x_\pm)$ leads to the Ragnisco–Bruschi Lax matrix $L(\rho)$. The last case corresponds to a deformation of the original Peakon Lax matrix in a direction orthogonal to the one chosen by Ragnisco–Bruschi.

We now establish the Poisson bracket structure between any two matrices of this family. From the Poisson brackets (2.11)–(2.17), we obtain

$$\left\{\bar{L}_1(a,\bar{a})\,,\,\bar{L}_2(b,\bar{b})\right\} = \frac{\nu}{4}\left[\bar{a}\Gamma_{21} - a(\Pi - \Pi^t)\,,\,\bar{L}_2(b,\bar{b})\right] - \frac{\nu}{4}\left[\bar{b}\Gamma_{12} - b(\Pi - \Pi^t)\,,\,\bar{L}_1(a,\bar{a})\right]$$

$$+ \frac{\nu}{4}\left[\Pi\,,\,(a\bar{b} + \bar{a}b)S_2 + (ab + \bar{a}\bar{b} - 1)A_2\right]$$

$$+ \frac{\nu}{4}\left[\Pi^t\,,\,-(a\bar{b} - \bar{a}b)S_2 + (\bar{a}\bar{b} - ab - 1)A_2\right]. \tag{4.17}$$

Since $\bar{L}(a,\bar{a})$ and $\bar{L}(b,\bar{b})$ are in the same family, $\frac{a^2 - \bar{a}^2 - 1}{\bar{a}} = \frac{b^2 - \bar{b}^2 - 1}{\bar{b}}$. Then one identifies consistently

$$\begin{cases} (a\bar{b} + \bar{a}b)S + (ab + \bar{a}\bar{b} - 1)A = c\big(\bar{L}(a,\bar{a}) - \bar{L}(b,\bar{b})\big) \\[2mm] -(a\bar{b} - \bar{a}b)S + (\bar{a}\bar{b} - ab - 1)A = c'\big(\bar{L}(a,\bar{a})^t - \bar{L}(b,\bar{b})\big) \end{cases} \quad \text{where} \quad \begin{cases} c = \dfrac{a\bar{b} + \bar{a}b}{\bar{a} - \bar{b}} \\[3mm] c' = -\dfrac{a\bar{b} - \bar{a}b}{\bar{a} - \bar{b}} \end{cases} \tag{4.18}$$

It leads to

$$\left\{\bar{L}_1(a,\bar{a})\,,\,\bar{L}_2(b,\bar{b})\right\} = \frac{\nu}{4}\left[\bar{a}\Gamma_{21} - a(\Pi - \Pi^t)\,,\,\bar{L}_2(b,\bar{b})\right] - \frac{\nu}{4}\left[\bar{b}\Gamma_{12} - b(\Pi - \Pi^t)\,,\,\bar{L}_1(a,\bar{a})\right]$$

$$+ \frac{\nu}{4}\,c\left[\Pi\,,\,\bar{L}_2(a,\bar{a}) - \bar{L}_2(b,\bar{b})\right] + \frac{\nu}{4}\,c'\left[\Pi^t\,,\,\bar{L}_2(a,\bar{a})^t - \bar{L}_2(b,\bar{b})\right]. \tag{4.19}$$

Finally, using once more the relations $\left[\Pi, M_2\right] = -\left[\Pi, M_1\right]$ and $\left[\Pi^t, M_2^t\right] = \left[\Pi^t, M_1\right]$ (for any matrix $M$), we can rewrite the above relation in a $r$-matrix form:

$$\left\{\bar{L}_1(a,\bar{a})\,,\,\bar{L}_2(b,\bar{b})\right\} = \left[\mathbf{r}_{12}(a,\bar{a};b,\bar{b})\,,\,\bar{L}_1(a,\bar{a})\right] - \left[\mathbf{r}_{21}(b,\bar{b};a,\bar{a})\,,\,\bar{L}_2(b,\bar{b})\right]$$

$$\mathbf{r}_{12}(a,\bar{a};b,\bar{b}) = \frac{\nu}{4}\left(-\bar{b}\,\Gamma_{12} + b\,(\Pi - \Pi^t) - c\,\Pi + c'\,\Pi^t\right), \tag{4.20}$$

$$\mathbf{r}_{21}(b,\bar{b};a,\bar{a}) = \frac{\nu}{4}\left(-\bar{a}\,\Gamma_{21} + a\,(\Pi - \Pi^t) + c\,\Pi + c'\,\Pi^t\right).$$

Using the values of $c$ and $c'$, one can consistently check that $c\big|_{(a,\bar{a})\leftrightarrow(b,\bar{b})} = -c$ and $c'\big|_{(a,\bar{a})\leftrightarrow(b,\bar{b})} = c'$, justifying the notation $\mathbf{r}_{21}(b,\bar{b};a,\bar{a})$ and $\mathbf{r}_{12}(a,\bar{a};b,\bar{b})$ in (4.20).

Let us stress that despite $\mathbf{r}$ apparently depends on four spectral parameters, there are only two independent ones on the complex variety $\frac{a^2 - \bar{a}^2 - 1}{\bar{a}} = \frac{\lambda^2 - \rho^2 - 1}{\rho}$, as it should be for an $r$-matrix. Indeed, one can parametrize the variety as

$$\bar{a} = \frac{1}{\alpha\,\sinh(z) - \frac{\gamma}{2}} \quad ; \quad a = \frac{\alpha\,\cosh(z)}{\alpha\,\sinh(z) - \frac{\gamma}{2}}, \quad z \in \mathbb{C}$$

$$\text{with} \quad \alpha^2 = 1 - \frac{\gamma^2}{4} \quad ; \quad \gamma = \frac{\lambda^2 - \rho^2 - 1}{\rho}. \tag{4.21}$$

Then, after a rescaling $\bar{L}(a,\bar{a}) \to \frac{1}{a}\bar{L}(a,\bar{a})$, we get an $r$-matrix of the form

$$\mathbf{r}_{12}(z_1;z_2) = \frac{\nu}{4}\left(-\Gamma_{12} + \frac{\sinh(z_1) + \sinh(z_2)}{\cosh(z_1) - \cosh(z_2)}\left(\alpha\sinh(z_2) - \frac{\gamma}{2}\right)\Pi\right.$$

$$\left. + \frac{\sinh(z_1) + \sinh(z_2)}{\cosh(z_1) + \cosh(z_2)}\left(\alpha\sinh(z_2) - \frac{\gamma}{2}\right)\Pi^t + \alpha\cosh(z_2)\left(\Pi - \Pi^t\right)\right). \tag{4.22}$$

Note also that the existence of a non-dynamical, spectral parameter dependent $r$-matrix is not inconsistent with the fact that the initial Lax matrix $\bar{L}(\lambda, \rho)$ in (2.19) does not possess a $r$-matrix. Indeed, in this case we have $a = b = \lambda$ and $\bar{a} = \bar{b} = \rho$. Then, since $\mathbf{r}(a, \bar{a}; b, \bar{b})$ is singular when $\bar{a} = \bar{b}$, the corresponding Poisson bracket has to be extracted by a non-trivial limiting procedure, breaking the $r$-matrix form.

**Conserved quantities without extended auxiliary space.** Now that we have expressed the Poisson brackets of $\bar{L}_1(a, \bar{a})$ and $\bar{L}_2(b, \bar{b})$ in the $r$-matrix form (4.20), standard arguments of $r$-matrix presentation show that the invariant quantities

$$\tau_n(a, \bar{a}) = \text{Tr}\big(\bar{L}(a, \bar{a})\big)^n \tag{4.23}$$

PB-commute:

$$\big\{\tau_n(a, \bar{a}), \, \tau_m(b, \bar{b})\big\} = 0. \tag{4.24}$$

Since $\tau_n\big(a(x), \bar{a}(x)\big) \equiv \tau_n(x)$, this ends the proof of property (4.13).

### Acknowledgments

J.A. wishes to warmly thank LAPTh for their kind hospitality and financial support. We wish to warmly thank the referees for their careful reading of the manuscript and their comments, which in particular contribute to the refinement of section 4.3.

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
