# Peer review of "A new integrable structure associated to the Camassa-Holm peakons"

_SciPost Physics_

## Round 1 · Referee Report · Anonymous (Referee 1) · 2023-10-20

Strengths

  1. Contains interesting new results on an important subject.
  2. Very clearly written.

Weaknesses

None

Report

This paper contains interesting new results concerning the peakon dynamics derived from the Camassa-Holm shallow-water equation. The peakons are soliton like, localized solutions exhibiting certain singularities. Their dynamics can be described in terms of a finite dimensional Hamiltonian system. The integrability of a one-parameter extension of the original N-peakon Hamiltonian system was established by Ragnisco and Bruschi in 1995, which forms the starting point of this manuscript.

The authors’ main result is that they found a new, two parameter generalization of the Ragnisco—Bruschi model, and gave an elegant proof of its integrability relying on a non-dynamical Poisson structure built on three N by N matrices whose linear combinations produces the Lax matrix.
Moreover, after adopting a certain restriction of the parameters (equation (2.32), which also occurs in the original Camassa—Holm case), they found new solutions of the Camassa—Holm equation that correspond to the deformation of the finite dimensional integrable Hamiltonian system of peakon dynamics. The paper contains further interesting results as well, such as a spectral parameter independent, non-dynamical r-matrix interpretation of the above mentioned Poisson algebra, as well as a spectral parameter dependent formulation of the Poisson structure.

The text is very clearly written. I noticed only one unclear statement: the sentence above equation (2.9) regarding the proof of integrability of the Ragnisco—Bruschi system, which will be easily corrected in proof.

In conclusion, I am happy to recommend the publication of this paper in SciPost.

---

## Round 1 · Referee Report · Anonymous (Referee 2) · 2023-10-23

Strengths

1- Interesting new results in the study of peakon dynamics and their integrable structures.

2- The paper exhibits some useful algebraic structures underlying these models.

3- The paper is well-written and overall easy to follow, even though some of the discussions are technical in nature.

Report

The main subject of this paper is the study of deformed $N$-peakon solutions of the Camassa–Holm wave equation and their integrable structure. Peakons are particular ansatzes for the field of the Camassa–Holm PDE which reduce the dynamics to a finite-dimensional system. A one-parameter integrable deformation of the standard peakon system was proposed by Ragnisco and Bruschi in 1995: its integrability relied on an implicit dynamical R-matrix. In this paper, the authors construct a further deformation of this system, now with two parameters, and show its integrability. To do so, they exhibit various explicit and interesting algebraic structures underlying the Poisson algebra of the theory, which notably do not rely on a dynamical R-matrix. Moreover, they show how this system can be obtained from a modified $N$-body ansatz in the Camassa–Holm PDE.

In my opinion, the results of this paper are interesting, original and bring new insights to the study of peakon solutions. Moreover, the article is well-written and overall easy to follow.

My main question on the paper concerns a rather precise and technical point, but which might have an important impact on some of the results (let me stress however that this does not call into question the overall viability of the paper, on the contrary). If I am not mistaken, the term $(a\bar{b}-\bar{a}b)S_2$ in (4.16) should in fact come with the opposite sign. From then on, using the identities (4.17), $[\Pi,M_1]=-[\Pi,M_2]$ and $[\Pi^t,M_1]=[\Pi^t,M_2^t]$, and after a few manipulations, it seems to me that one can obtain a slightly modified equation (4.18) with all matrices $\bar{L}(a,\bar{a})$ on the first tensor factor and all matrices $\bar{L}(b,\bar{b})$ on the second one. Remarkably, the final result would then take the form of a R-bracket, with a spectral parameter dependent non-dynamical R-matrix $R(a,\bar{a},b,\bar{b})$. Note that this is not inconsistent with the fact that the initial Lax matrix $\bar{L}=\bar{L}(\lambda,\rho)$ in (2.19) does not possess a R-bracket since $R(a,\bar{a},b,\bar{b})$ would be singular when $\bar{a}=\bar{b}$: the bracket for $(a,\bar{a})=(b,\bar{b})=(\lambda,\rho)$ would then be extracted by a non-trivial limiting procedure, breaking the R-matrix form. Although very minor, this modification in (4.16) would then result in an even simpler and more apparent integrable structure. This should of course be carefully checked and analysed by the authors. If the above is correct, the corresponding equations should be modified in the manuscript and it seems to me that it would be useful to include and discuss the R-matrix interpretation in the paper.

Beyond this main point, I list in the section "Requested changes" a few typos I have spotted as well as some minor suggestions, which the authors should feel free to follow or not. To conclude, I think the paper presents interesting new results and is well-written and I will be happy to recommend it for publication in SciPost Physics once the point raised above is unravelled.

Requested changes

1- Mainly, I would like to ask the authors to clarify the point raised in the report above about equations (4.16)-(4.18) and if necessary, to implement the relevant modifications on the paper.

Typos:

2- Before (2.9), the text refers to an unlabelled appendix.

3- Technically, I think that there should be an additional $1/n$ factor in the first equality of (2.24) (although this is in the end inconsequential).

4- After equation (4.15), it seems to me that the choice $x=\frac{\lambda-1-3\rho}{\lambda-1+\rho}$ leads to $(a,\bar{a})=(-1,0)$ and thus sends $\bar{L}(a,\bar{a})$ to $T-A$, while the initial Lax matrix (2.19) is recovered from $x=1$.

5- In (4.16) and subsequent equations, should there be a global factor $\nu/8$?

6- In the second line of (4.17), there is a tensor index $2$ left on the matrix $S$.

Minor comments and suggestions, which the authors should feel free to address/implement or not:

7- Although the paper is overall quite easy to follow, the section 2.3 is rather technical and in my (subjective) opinion, it could help the less familiar readers to add some minor additional explanations throughout the reasoning. I list some potential suggestions below: - mentioning $\Pi^2=\mathbb{I}_{N^2}$ and $\text{Tr}_2(\Pi)=\mathbb{I}_N$ - mentioning early on that $T$ and $S$ are symmetric and $A$ is anti-symmetric - the properties $[\Pi,M_1]=-[\Pi,M_2]$ and $[\Pi^t,M_1]=[\Pi^t,M_2^t]$ (this is essentially (2.26) but I think that this property, combined with the one above, is used earlier from (2.20) to (2.23))

8- It could be worth commenting on how restrictive the choice (2.32) is for the interpretation of the deformed system as a modified peakon CH solution, if their is a simple enough answer to that question (in particular, does the case with arbitrary $\nu$ come from a simple modification of the CH equation, for instance by rescaling $t$, $x$ and $u$?)

---

## Editorial Decision

resubmitted